# An Experimental Investigation of Multiple Sluicing in Mandarin Chinese

Xue Bai [1] , Álvaro Cortés Rodríguez [2,3,*] and Daiko Takahashi [1]

1   Graduate School of International Cultural Studies, Tohoku University, Sendai 980-8576, Japan;
    bai.xue.t6@dc.tohoku.ac.jp (X.B.); daiko@tohoku.ac.jp (D.T.)
2   Faculty of Humanities, University of Tübingen, 72074 Tübingen, Germany
3   Department of English and American Studies, Faculty of Humanities, University of Kassel,
    34125 Kassel, Germany
*   Correspondence: alvaro.cortes-rodriguez@uni-tuebingen.de

**Abstract:** This paper examines multiple sluicing constructions in Mandarin Chinese (henceforth, MC) experimentally. The acceptability status of such constructions in MC is controversial, and the judgments reported in the previous literature vary. Obtaining experimental evidence on the acceptability status is, therefore, important to advance the research on multiple sluicing in MC. Consequently, the present study conducts two sets of experiments to investigate factors affecting the acceptability of multiple sluicing sentences and the influence of the distribution of *shi* preceding *wh*-remnants on acceptability ratings. The results show that multiple sluicing in MC is generally a marked construction. Nevertheless, factors including prepositionhood and specificity have ameliorating effects on the acceptability of such constructions. Moreover, the influence of the distribution of *shi* on the acceptability ratings is related to the nature of *wh*-remnants; that is, its presence significantly improves the acceptability of cases of multiple sluicing when it precedes bare *wh*-arguments. We argue that the observed ameliorating effects on multiple sluicing can be explained by a cue-based retrieval approach to cross-linguistic elliptical constructions. Compared to bare *wh*-arguments, prepositional and discourse-linked *wh*-phrases provide cues to facilitate the retrieval of information from antecedent clauses.

**Keywords:** multiple sluicing; Mandarin Chinese; experimental syntax

## 1. Introduction

Coined by Ross (1969), sluicing is the ellipsis process by which questions like (1a) are converted into reduced forms like (1b).

(1)  a.   He is writing something, but you can't imagine [what he is writing].
     b.   He is writing something, but you can't imagine [what].
          (Ross 1969, p. 252)
     c.   He is writing something, but you can't imagine [$_{CP}$ what$_i$ [$_{TP}$ he is writing $t_i$] ]

The embedded clause in (1a), indicated by the brackets, contains a *wh*-question, which is reduced to only contain a *wh*-phrase in (1b). The full-fledged *wh*-question and the reduced *wh*-question have the same interpretation (e.g., Ross 1969; Lasnik 2001; Merchant 2001). The remaining *wh*-phrase in (1b), namely, *what*, is called a *wh*-remnant, which has a corresponding counterpart in the preceding clause, i.e., *something*, called a correlate. As discussed in the previous literature (e.g., Ross 1969; Merchant 2001), the sluicing sentence in (1b) is derived by moving the *wh*-phrase *what* into the specifier position of CP, which is followed by TP-ellipsis, indicated with grey shading, as illustrated in (1c).

The type of sluicing configuration containing one *wh*-remnant, as in (1b), is called single sluicing. Sluicing, however, also allows multiple remnants, a configuration known as multiple sluicing (Takahashi 1994). An example is shown in (2).[1]

(2) ?  Everybody brought something (different) to the potluck, but I couldn't tell you who what.
(Merchant 2001, p. 112)

(2) has two *wh*-remnants, *who* and *what*. In this paper, we focus on multiple sluicing in Mandarin Chinese (hereafter, MC). Consider the example below:[2]

(3)

| Mouren | tou-le | tade | yi | yang | dongxi, | wo | xiang | zhidao | *(shi) |
|--------|--------|------|-----|------|---------|-----|-------|--------|--------|
| someone | steal-PFV | his | one | CLF | thing | I | want | know | SHI |
| shei | *(shi) | shenme. | | | | | | | |
| who | SHI | what | | | | | | | |

'lit. Someone stole one of his belongings, and I wonder who what.'
(Adams and Tomioka 2012, p. 237)

The first clause in (3) functions as the antecedent for the sluiced clause containing two bare *wh*-arguments, *shei* 'who' and *shenme* 'what,' both accompanied by *shi*, a copula in MC.

It is worth noting the particular usage of the term *multiple sluicing* we are following in this paper. Sluicing and multiple sluicing generally refer to the ellipsis process of movement of *wh*-remnants followed by ellipsis of TP, as illustrated in (1c) in English. In some languages like MC, the theoretical analysis of truncated indirect questions like (3) has been under debate. In MC, three approaches have been discussed in the previous literature: the movement-and-deletion analysis (e.g., Chiu 2007; Bai and Takahashi 2023), the pseudo-sluicing analysis (e.g., Adams and Tomioka 2012), and the combined analysis of the former two analyses (e.g., Wang and Han 2018). The detailed derivational process of truncated questions in MC is not the concern of the present paper. Therefore, *multiple sluicing* in this paper is used as a cover term referring to the phenomenon of reduced indirect questions in MC without committing to a specific syntactic derivation. Yet, in Section 4.2, we show that our experimental results do not seem to favor the pseudo-sluicing analysis.

Multiple sluicing in MC has been studied for over a decade, but its acceptability status is still debated (Chiu 2007; Adams and Tomioka 2012; Takahashi and Lin 2012; Park and Li 2013; Wang 2018; Wang and Han 2018; Bai and Takahashi 2023). Some linguists (Chiu 2007; Wang 2018) consider cases like (3) unacceptable, while others (Adams and Tomioka 2012; Park and Li 2013; Wang and Han 2018) make the opposite judgment. In addition, the reported judgments in the previous literature are elicited based on informal data collection. Experimental evidence on the acceptability of multiple sluicing under formal experimental settings is, therefore, essential for the in-depth investigation of this phenomenon in MC.

The purpose of this paper is twofold. It aims first to support the cue-retrieval analysis, which accounts for cross-linguistic multiple sluicing constructions (Cortés Rodríguez 2023), by adding another language, namely MC, into the dataset. The other purpose of the present study is to present experimental evidence to further the discussion on multiple sluicing in MC, particularly regarding the distribution of *shi*. In this paper, we report the results of two experiments. The first experiment was modeled after the cross-linguistic experiments on multiple sluicing conducted by Cortés Rodríguez (2023). The second was designed to examine the effect of the distribution of *shi* on the acceptability of multiple sluicing.

The remainder of this paper is organized as follows. Section 2 presents debates on multiple sluicing in MC. Additionally, factors influencing the acceptability of multiple sluicing constructions cross-linguistically, such as prepositionhood and specificity, are reviewed. Section 3 details two acceptability judgment studies on multiple sluicing in MC. The first one investigates the influence of prepositionhood and specificity on the acceptability of such constructions. The second one examines how the distribution of *shi* affects acceptability ratings. Section 4 discusses the experimental results. Finally, Section 5 concludes this paper.

## 2. Background

### 2.1. The Debates on Multiple Sluicing in Mandarin Chinese

This section details the debates on multiple sluicing in MC. Let us start by describing single sluicing in the language. Consider examples (4) and (5):

(4) | Zhangsan | kan-dao | mouren, | danshi | wo | bu | zhidao | *(shi) | shei. |
| Zhangsan | see-PFV | someone | but | I | not | know | SHI | who |

'Zhangsan saw somebody, but I don't know who.'
(Wei 2004, p. 165)

(5) | Zhangsan | mai-le | mouwu, | danshi | wo | bu | zhidao | *(shi) | shenme. |
| Zhangsan | buy-PFV | something | but | I | not | know | SHI | what |

'Zhangsan bought something, but I don't know what.'
(ibid.)

The *wh*-remnants in the sluiced clauses in (4) and (5) are simplex/bare *wh*-arguments *shei* 'who' and *shenme* 'what,' respectively, which must be accompanied by *shi*, as discussed in the previous literature (Wang 2002; Adams 2004; Wei 2004; Wang and Wu 2006; Chiu et al. 2008; Park and Li 2013; Li and Wei 2014; 2017; Song 2016; Sun 2018; Zhang and Overfelt 2019; Lee 2020).

The status of *shi* in sluicing in MC has received much debate. While some linguists (Wang 2002; Wang and Wu 2006; Qin and Xu 2019) regard it as a kind of focus marker, others (Adams 2004; Adams and Tomioka 2012; Li and Wei 2017) claim that it is a copula. Both claims are reasonable given that *shi* in the language can function as a focus marker and as a copula, as shown in (6) and (7), respectively.

(6) | Shi | wo | mingtian | cheng | huoche | qu | Guangzhou. |
| FOC | I | tomorrow | ride | train | go | Guangzhou |

'It is I who will go to Guangzhou by train tomorrow.'
(Xu 2003, p. 4)

(7) | Ta | shi | yi | ge | xuesheng. |
| he | COP | one | CLF | student |

'He is a student.'
(ibid.)

The claim that *shi* is a focus marker in reduced questions is used to support the movement-and-deletion analysis (e.g., Wang and Wu 2006). On the other hand, the claim that *shi* is a copula in reduced questions is employed to support the pseudo-sluicing analysis. Since we do not discuss the derivational process of truncated questions in MC, the specific status of *shi* is not the main concern of this paper. We will only discuss the usage and distribution of *shi* in reduced questions in Section 4.

In addition to bare *wh*-arguments, sluicing in MC allows specific *wh*-arguments, as illustrated in (8).

(8) | Lisi | bu | xihuan | yi | shou | ge, | danshi | wo | bu | zhidao | (shi) |
| Lisi | not | like | one | CLF | song | but | I | not | know | SHI |
| na | yi | shou | ge. |
| which | one | CLF | song |

'Lisi doesn't like one song, but I don't know which song.'
(Adams and Tomioka 2012, p. 223)

In the sluiced clause, the specific *wh*-argument *nayishou ge* 'which song' can be optionally accompanied by *shi* (Adams 2004; Wei 2004; Adams and Tomioka 2012; Park and Li 2013; Li and Wei 2014; 2017; Song 2016; Sun 2018; Zhang and Overfelt 2019).

Sluicing in MC also allows adjunct and prepositional *wh*-remnants, as in (9) and (10), respectively.

(9)  Zhangsan zai  mou  ge  difang  chu  shi  le,  danshi  wo  bu
     Zhangsan at  some  CLF  place  have  accident  PRF  but  I  not
     zhidao (shi) zai  nali.
     know  SHI  at  where
     'Zhangsan had an accident at some place, but I don't know where.'
     (Wei 2004, p. 168)

(10) Zhangsan gang  gen  mouren  likai-le,  danshi  wo  bu  zhidao  (shi)  gen/han  shei.
     Zhangsan just  PREP  someone  leave-PFV  but  I  not  know  SHI  PREP  who
     'Zhangsan just left with someone, but I don't know with whom.'
     (ibid.)

As discussed in the literature, the occurrence of *shi* is optional in front of adjunct and prepositional *wh*-remnants (Adams 2004; Wei 2004; Wang and Wu 2006; Park and Li 2013; Song 2016; Zhang and Overfelt 2019; Lee 2020). Moreover, Adams (2004) mentions that these single sluicing sentences are more natural when *shi* appears with *wh*-remnants. Thus far, we can see that the presence of *shi* is obligatory with bare *wh*-arguments but optional with specific *wh*-arguments, *wh*-adjuncts, and prepositional *wh*-remnants.[3]

In addition to single sluicing, multiple sluicing is found in MC. Let us start our discussion with *wh*-arguments. Chiu (2007) states that cases of multiple sluicing with two *wh*-arguments are unacceptable, as illustrated in (11).

(11) * Mouren  da-le  women  ban  de  ren,  dan  wo  bu  zhidao  *(shi)  shei  shei.
       someone  hit-PFV  our  class  GEN  person  but  I  not  know  SHI  who  who
       'Someone hit a person of our class, but I don't know who whom.'
       (Chiu 2007, p. 23)

As discussed in Adams and Tomioka (2012) and Wang and Han (2018), cases like (11) are indeed unacceptable. The unacceptability may be caused by the presence of two identical *wh*-remnants whose correlates in the first clause cannot be identified. The *wh*-remnants may not, therefore, be able to be properly interpreted. Similar cases of multiple sluicing in English are also judged unacceptable (e.g., Bolinger 1978; Richards 2010), as shown in (12).

(12) * I know that in each instance one of the girls chose one of the boys. But which which?
       (Bolinger 1978, p. 109)

Identical *wh*-remnants are claimed to cause a homonymic conflict, which renders the relevant multiple sluicing sentences unacceptable (Bolinger 1978).

Adams and Tomioka (2012), on the other hand, observe that multiple sluicing with two different *wh*-arguments is allowed, as shown above in (3) and repeated below as (13).

(13) Mouren  tou-le  tade  yi  yang  dongxi,  wo  xiang  zhidao  *(shi)
     someone  steal-PFV  his  one  CLF  thing  I  want  know  SHI
     shei  *(shi)  shenme.
     who  SHI  what
     'lit. Someone stole one of his belongings, and I wonder who what.'
     (Adams and Tomioka 2012, p. 237)

In (13), the two *wh*-arguments are *shei* 'who' and *shenme* 'what'. Park and Li (2013) and Wang and Han (2018) agree with Adams and Tomioka (2012) that cases like (13) are acceptable. Wang (2018), on the other hand, rejects such cases. See the example (14) below provided by Wang (2018).

(14) * Lisi  zhi  jide  you  ren  mai-le  dongxi,  dan  ta  wang-le  shi
       Lisi  only  remember  have  person  buy-PFV  thing,  but  he  forget-PFV  SHI
       shenme  (shi)  shei.
       what  SHI  who
       'Lisi only remembered someone bought something, but he forgot what who.'
       (Wang 2018, p. 1)

Although we agree with the judgment provided for (14), we suspect that rather than by the combination of two bare *wh*-arguments, the reported unacceptability is caused by the following two factors. First, the order of the *wh*-arguments in (14) should be reversed because multiple sluicing is reported to adhere to the superiority effect, according to which the order of *wh*-remnants should conform to that of their correlates (e.g., Merchant 2001; Adams and Tomioka 2012; Kotek and Barros 2018). Moreover, the second *shi* preceding the bare *wh*-argument, *shenme* 'what,' must be obligatory, based on discussions in the previous literature (Adams and Tomioka 2012; Park and Li 2013; Wang and Han 2018), as well as on the results of our exploratory test.[4] The obligatory presence of *shi* in front of each bare *wh*-argument is not surprising since *shi* is also obligatory in front of bare *wh*-arguments in single sluicing in MC.

Multiple sluicing with two specific *wh*-arguments is allowed, as discussed in Wang and Han (2018). Consider the example in (15). Note that neither of the specific *wh*-arguments is preceded by *shi*.

(15)　Mouren　mai-le　yi　yang　dongxi,　danshi　wo　bu　zhidao　na　ge　ren
　　　 someone　buy-PFV　one　CLF　thing　but　I　not　know　which　CLF　person
　　　 na　yang　dongxi.
　　　 which　CLF　thing
'lit. Someone bought something, but I don't know which person which thing.'
(Wang and Han 2018, p. 611)

Moreover, cases of multiple sluicing with a *wh*-argument and a *wh*-adjunct are also allowed (Chiu 2007; Adams and Tomioka 2012). Consider the examples below:

(16)　Mouren　da-le　women　ban　de　ren,　dan　wo　bu　zhidao　*(shi)　shei　zai
　　　 someone　hit-PFV　our　class　GEN　person　but　I　not　know　SHI　who　at
　　　 nali.
　　　 where
'lit. Someone hit a person of our class, but I don't know who where.'
(Chiu 2007, p. 23)

(17)　Laoshi　chufa-le　mouren,　wo　xiang　zhidao　*(shi)　shei
　　　 teacher　punish-PFV　someone　I　want　know　SHI　who
　　　 (shi)　wei　shenme.
　　　 SHI　for　what
'lit. Teacher punished someone, and I wonder who why.'
(Adams and Tomioka 2012, p. 237)

Adams and Tomioka (2012) and Park and Li (2013) mention that the occurrence of *shi* preceding *wh*-adjuncts is optional, which is also the case for single sluicing in MC.[5] Moreover, according to Adams and Tomioka (2012), (17) is more natural if *shi* accompanies each *wh*-remnant.

Likewise, Wang and Han (2018) discuss multiple sluicing with a specific *wh*-argument and a *wh*-adjunct, as illustrated in (18).

(18)　Mouren　zai　mou　ge　difang　mai-le　yi　jian　chenyi,　danshi　wo
　　　 someone　at　some　CLF　place　buy-PFV　one　CLF　shirt　but　I
　　　 bu　zhidao　na　ge　ren　zai　nali.
　　　 not　know　which　CLF　person　at　where
'lit. Someone bought a shirt at a certain place, but I don't know which person where.'
(Wang and Han 2018, p. 611)

It is worth mentioning that neither of the *wh*-remnants in (18) is preceded by *shi*. Additionally, Adams and Tomioka (2012) provide examples with a complex *wh*-argument and a *wh*-adjunct, as in (19).

(19)　Zhangsan　zai　　　　moushi　qu　　　mai-le　　yi　　　yang　ta　　　hen　　xihuan　de
　　　　Zhangsan　at　　　　sometime　go　　buy-PFV　one　　CLF　　he　　　very　　like　　GEN
　　　　dongxi,　　danshi　　wo　　　bu　　　zhidao　　(shi)　zai　　heshi　*(shi)　shenme　dongxi.
　　　　thing　　　but　　　I　　　not　　　know　　　SHI　　at　　　when　　SHI　　what　　thing
　　　　'lit. Zhangsan went to buy something he really liked at some time, but I don't know when what thing.'
　　　　([Adams and Tomioka 2012], p. 239)

Here we can see that the authors regard the presence of *shi* as optional with the *wh*-adjunct but obligatory with the complex *wh*-argument.

In this section, we have reviewed the debates on multiple sluicing in MC involving different combinations of *wh*-remnants.[6] First, varied judgments on the acceptability of multiple sluicing, such as cases involving two bare *wh*-arguments, have been reported. Next, research on cases with two specific *wh*-arguments has been shown to be lacking. Furthermore, the reported judgments in the previous literature were elicited based on informal data collection. Lastly, we have reiterated reports from the previous literature where various distributions of *shi* depending on the nature of *wh*-phrases have been presented. Thus far, no studies have provided a comprehensive discussion on the influence of the different distributions of *shi* on the acceptability of multiple sluicing sentences.

### 2.2. Aspects Affecting the Acceptability of Multiple Sluicing Cross-Linguistically

### 2.2.1. The Presence of a Preposition

Multiple sluicing in English has also been studied in the previous literature ([Takahashi 1994]; [Nishigauchi 1998]; [Merchant 2001], [2006]; [Fox and Pesetsky 2003]; [Richards 2010]; [Hoyt and Teodorescu 2012]; [Takahashi and Lin 2012]; [Lasnik 2014]; [Barros and Frank 2016], [2022]; [Abels and Dayal 2017], [2022]; [Kotek and Barros 2018]; [Cortés Rodríguez 2022], [2023]). As in MC, the acceptability of multiple sluicing sentences in English is also under debate. Some linguists mention that cases with two *wh*-arguments are degraded or unacceptable, as shown in (20). On the other hand, cases with a *wh*-argument and a prepositional *wh*-remnant are more acceptable ([Fox and Pesetsky 2003]; [Richards 2010]; [Lasnik 2014]), as in (21).

(20)　?*　　Someone saw something, but I can't remember who what.
　　　　　　([Lasnik 2014], p. 8)

(21)　?　　Someone talked about something, but I can't remember who about what.
　　　　　　(ibid.)

Note that the correlates of the *wh*-remnants in (20)–(21) are existential quantifiers, namely, *someone* and *something*. The previous literature also discusses cases where the first correlate is a universal quantifier and the second correlate is an existential quantifier ([Bolinger 1978]; [Nishigauchi 1998]; [Merchant 2001], [2006]; [Richards 2010]). Consider the following examples:

(22)　*　　I know every man insulted a woman, but I don't know which man which woman.
　　　　　　([Richards 2010], p. 3)

(23)　　　I know every man danced with a woman, but I don't know which man with which woman.
　　　　　　(ibid.)

[Richards] ([2010]) observes that while the multiple-sluicing sentence in (22) with two *wh*-arguments is not acceptable, the sentence in (23) with a *wh*-argument and a prepositional *wh*-remnant is acceptable. Nevertheless, some linguists (e.g., [Bolinger 1978]; [Nishigauchi 1998]; [Merchant 2001], [2006]; [Barros and Frank 2016]; [Kotek and Barros 2018]) mention that cases of multiple sluicing with two *wh*-arguments are acceptable or only mildly deviant when the first correlate is a universal quantifier, as shown in examples (2) and (24).

(24)　?　　Everyone bought something, but I couldn't tell you who what.
　　　　　　([Merchant 2006], p. 284)

So far, we have briefly reviewed the varied judgments on multiple sluicing in English. Since the judgments reported in the previous literature are elicited through informal tests,

Cortés Rodríguez (2023) initiates formal experimental studies to examine the acceptability of such constructions and the factors that could influence their acceptability. One of the tested factors is the presence of a prepositional *wh*-remnant as the second remnant in multiple sluicing. See the test items from Cortés Rodríguez (2023) in (25)–(26).

(25)　　　Everyone completed something, but I just don't know who what.
(26)　　　Everyone commented on something, but I just don't know who on what.
　　　　(Cortés Rodríguez 2023, p. 7)

The multiple-sluiced clause in (25) has two *wh*-arguments, and that in (26) has a *wh*-argument and a prepositional *wh*-remnant. Cortés Rodríguez (2023) conducts three experiments whose results demonstrate that cases with a *wh*-argument and a prepositional *wh*-remnant such as (26) are significantly more acceptable than those with two *wh*-arguments such as (25). The author concludes that the presence of a preposition is an ameliorating factor in improving the acceptability of multiple sluicing in English, which confirms the observations made in the previous literature (Richards 2010; Lasnik 2014).

This ameliorating effect is explained following a cue-based retrieval approach to ellipsis (e.g., Martin and McElree 2008, 2011; Nykiel et al. 2023). Understanding a sentence requires retrieving information from working memory. During the retrieving process, the syntactic and semantic information that enables direct access to relevant memory representations is called cues (McElree et al. 2003; Lewis et al. 2006). As discussed in the previous literature (Lewis and Vasishth 2005; Harris 2015, 2019; Cortés Rodríguez 2023), morphosyntactic and lexical features, such as morphological case, gender, plurality, nominal restrictors, and prepositions, constitute cues for the retrieval of information. In this respect, cues serve to identify a previously stored linguistic item and to disambiguate it from other interfering items. Information retrieval is carried out via matching stored items with retrieval cues (Van Dyke and Lewis 2003). The cue-base retrieval modal has been applied to explain many sentence constructions like relative clauses and elliptical constructions, such as sluicing and multiple sluicing constructions. Elliptical constructions are generally preceded by linguistic antecedents. Cues facilitate the information retrieval from antecedents and, as a result, facilitate the processing of elliptical constructions (e.g., Martin and McElree 2008, 2011; Nykiel et al. 2023). As discussed in Cortés Rodríguez (2023), prepositions function as a cue for identifying the argument-structural status/thematic role of a sluiced *wh*-phrase within the inferred proposition, i.e., whether the *wh*-phrase is understood as associated with a subject, or a direct or indirect object (or also other phrases associated with the verbal event of the inferred proposition). Prepositions facilitate the retrieval of information presented in antecedent clauses and the identification of the correlate–remnant relation. Two nominal *wh*-arguments, on the other hand, increase the difficulty of discerning which correlate a *wh*-remnant refers to and, as such, impose a processing burden because of containing fewer cues. Consequently, factors that facilitate processing lead to higher acceptability ratings.

### 2.2.2. Specificity

As discussed in Section 2.2.1, some cases of multiple sluicing in English are reported to be degraded, as in (20), repeated below as (27). Lasnik (2014) mentions that such cases become less degraded when the second remnant becomes "heavier", as illustrated in (28).

(27)　?*　　Someone saw something, but I can't remember who what.
　　　　　(Lasnik 2014, p. 8)
(28)　?　　Some linguist criticized (yesterday) some paper about sluicing, but I don't know which linguist which paper about sluicing.
　　　　　(ibid.)

In (28), the second remnant *which paper about sluicing*, containing *which* plus a complex NP restrictor *paper about sluicing*, is heavier than the bare *wh*-argument *what* in (27). According to Lasnik (2014), cases like (28) with heavy remnants are less degraded than cases like (27).

Since Lasnik (2014) is the only study that mentions the influence of the weight of *wh*-remnants on the acceptability ratings of multiple sluicing in English, Cortés Rodríguez (2023) conducts three experiments to examine whether the weight of the second *wh*-remnant influences the acceptability of such constructions. See (29) for a test item from Cortés Rodríguez (2023).

(29)    a.    Everyone completed something, but I just don't know who what.
        b.    Everyone completed some essay, but I just don't know who which essay.
        c.    Everyone completed some essay about colonialism, but I just don't know who which essay about colonialism.
        (Cortés Rodríguez 2023, p. 7)

In Cortés Rodríguez (2023), three levels of weight were examined: bare *wh*-arguments in (29a), specific *wh*-arguments in (29b), and heavy *wh*-arguments in (29c). The overall experimental results showed no significant difference in acceptability between the levels *bare* and *specific*. However, there was a detrimental effect between *bare/explicit* and *heavy*: the acceptability ratings got lower as the second remnants got heavier. Cortés Rodríguez (2023) argues that heavy remnants include repeated information from antecedent clauses, which causes the lowering in acceptability ratings. Similar observations have been made for single sluicing; that is, the more repeated material, the lower the acceptability rating (Gordon et al. 1993; Sag and Nykiel 2011).

On the other hand, Bhattacharya and Simpson (2012) mention that some cases of sluicing in Hindi with specific *wh*-arguments are judged more acceptable than those with bare *wh*-arguments. They argue that this difference can be attributed to the nature of *wh*-remnants. Specific *wh*-arguments provide unambiguous clues for establishing the correlate–remnant matching relation. Bare *wh*-arguments, on the other hand, cause parsing difficulties. Furthermore, Harris (2015) conducts eye-tracking studies on single sluicing in English, and the results support the cue-based parsing model of sentence processing. Concretely, discourse-linked *wh*-phrases containing *which* and nominal restrictors, such as *which wines*, provide richer cues than *wh*-phrases such as *which ones* with fewer cues. The former cases are proven to be able to facilitate the correlate–remnant pairing in sluicing and the accurate retrieval of information from antecedent clauses (see Harris 2015 for details; see also Harris 2019).

Based on the discussions in this section, we can see that there are discrepancies in the literature with respect to the effects of specificity in multiple sluicing configurations. For this reason, we were motivated to examine whether specificity is an ameliorating factor in multiple sluicing in MC.

### 2.2.3. The Cue-Based Retrieval Approach to Ellipsis

The cue-based retrieval approach to ellipsis (Martin and McElree 2008, 2011; Harris 2015, 2019; Nykiel et al. 2023) is supported by the experimental results in Cortés Rodríguez (2023). Despite multiple sluicing in English being a marked construction whose acceptability rating is in the 4 range on a 7-point Likert scale (as per the results in Cortés Rodríguez 2023), there are ameliorating effects contributed by some factors such as prepositionhood (i.e., the presence of a preposition accompanying the non-initial *wh*-remnant). Furthermore, Cortés Rodríguez (2023) provides additional evidence to support the cue-based analysis by conducting experiments on multiple sluicing in Spanish and German (Cortés Rodríguez 2021, 2023). In Spanish, a language with poor case morphology, the prediction for multiple sluicing is similar to that in English: The preposition effect should be observed since the cues provided by prepositions can facilitate the correlate–remnant pairing when no morphological case can provide cues. The experiment results in Cortés Rodríguez (2021) show that this prediction is indeed borne out. Moreover, similar to English, multiple sluicing in Spanish is also a marked construction. On the other hand, in German, a language with rich case morphology, multiple sluicing is predicted to be more acceptable than in English and Spanish. In addition, since case morphology provides sufficient cues for processing, the preposition effect is predicted not to be observed in German. These predictions are also borne out (see Cortés Rodríguez 2023).

Cues provided by syntactic and semantic features are proven to facilitate the processing of cross-linguistic elliptical constructions, though the strength of the cues in helping information retrieval may vary depending on language-specific properties. Inspired by prior research, we examine whether the cue-retrieval analysis can be supported by multiple sluicing in another language, namely, MC, following the experimental design of Cortés Rodríguez (2023). Specifically, we aim to investigate whether cues provided by prepositions and specific *wh*-arguments are effective in helping the processing of multiple sluicing in MC.

## 3. Experiments on Multiple Sluicing in Mandarin Chinese

This section presents two sets of experiments on multiple sluicing in MC. The first one investigates the influence of the presence of prepositions and specific *wh*-remnants on the acceptability of multiple sluicing. The second one contains a series of sub-experiments examining the different distributions of *shi*.

### 3.1. Experiment 1: Prepositionhood and Specificity

3.1.1. Methods

Design and Materials

We conducted an acceptability judgment experiment to examine the influence of the factors PREPOSITIONHOOD and SPECIFICITY on the acceptability of multiple sluicing in MC, following the experimental design of Cortés Rodríguez (2023). Twenty-four sentence quadruplets were created containing multiple sluicing using a $2 \times 2$ within-subject design. The two independent variables were (i) PREPOSITIONHOOD, representing the levels *+P* (presence of a preposition) and *-P* (absence of a preposition), and (ii) SPECIFICITY, including the factor levels *bare* (*wh*-pronoun) and *specific* (which NP).

Each experimental sentence consists of three parts: antecedent, *intro*, and sluice. The antecedent encompasses a universal quantifier and an existential quantifier. Each of the initial correlates is an animate entity, and each of the second correlates is an inanimate entity. Second, the *intro* part is the governing expression, *wo zhishi bu zhidao* 'I just don't know,' which selects an embedded clausal question. Lastly, the sluice part presents two adjacent *wh*-phrases in an elliptical context. Each test sentence displays harmony between the correlate and its corresponding sluiced *wh*-remnant. Here, harmony refers to equal weighting in each correlate–remnant pair. For instance, when the correlate is a complex phrase such as *some student*, the remnant is also a complex phrase such as *which student*. The test sentences were constructed in this manner to avoid the possible detrimental effect of disharmony on acceptability judgments (see Dayal and Schwarzschild 2010; Nykiel 2013). Furthermore, congruence is obtained between the two *wh*-remnants in each test item, meaning that the second remnant is in accordance with the first remnant with respect to specificity. Concretely, when the first remnant is a bare *wh*-phrase, the second remnant is also a bare *wh*-phrase (e.g., *who what*), and when the first remnant is a specific *wh*-phrase, the second remnant is also a specific *wh*-phrase (e.g., *which student which project*). Congruence is a factor examined in Cortés Rodríguez (2023), which is shown to affect acceptability, i.e., the acceptability is lower when there is incongruence between the two remnants.

Before presenting test items in the experiment, we note one important point with respect to the items. As reported in Section 2.1, multiple sluicing in MC shows various distributions of *shi* depending on the nature of the *wh*-phrases. Including all the distributions is beyond the scope of this experiment. To eliminate the influence of *shi* on experimental results, we used the most acceptable distribution (to our knowledge) in each condition. Section 3.2 will present an experiment series on the different distributions of *shi*, whose results demonstrate that the distribution of *shi* employed here is justified. Thus, the results obtained for Experiment 1 are not influenced by *shi,* and the obtained results are the product of the experimental manipulations.

Next, the items in each condition are explained. In the -P/bare condition, we used multiple sluicing sentences with *shi* preceding each *wh*-argument because the presence of

*shi* is obligatory with each bare *wh*-argument (Adams and Tomioka 2012; Park and Li 2013; Wang and Han 2018), as shown in (30). In the -P/specific condition, we employed multiple sluicing sentences with *shi* preceding only the first *wh*-argument because cases with such distribution were more acceptable than those with *shi* preceding each specific *wh*-argument, according to the results of our exploratory tests.[7] See (31) below for an illustration.

(30) **Condition 1: -P/bare**

| Mei | ge | ren | dou | wancheng-le | moushi, | wo | zhishi | bu | zhidao |
|-----|-----|--------|-----|-------------|-----------|-----|--------|-----|--------|
| every | CLF | person | all | complete-PFV | something | I | just | not | know |

| shi | shei | shi | shenme. |
|-----|------|-----|---------|
| **SHI** | who | **SHI** | what. |

'Everyone completed something, I just don't know who what.'

(31) **Condition 2: -P/specific**

| Mei | ge | daxuesheng | dou | wancheng-le | yi | ge | xiangmu, | wo |
|-----|-----|----------------|-----|-------------|-----|-----|----------|-----|
| every | CLF | college.student | all | complete-PFV | one | CLF | project | I |

| zhishi | bu | zhidao | shi | na | ge | daxuesheng | na | ge | xiangmu. |
|--------|-----|--------|-----|-----|-----|----------------|-----|-----|----------|
| just | not | know | **SHI** | which | CLF | college.student | which | CLF | project. |

'Every college student completed a project, I just don't know which college student which project.'

Moving on to the items containing a preposition, namely, +P conditions, we must recall that in MC, prepositional phrases usually precede verbs (Li and Thompson 1981; Yuan 2010; Ross and Ma 2014; Liu et al. 2019), as shown in the examples below.[8]

(32)

| Lao | changzhang | zhengzai | gei | linzi | li | de | shu | jiaoshui. |
|-----|------------|----------|------|-------|--------|------|------|-----------|
| old | farm.leader | now | **PREP** | woods | inside | GEN | tree | water |

'The farm leader is watering the trees in the woods.'
(Liu et al. 2019, p. 290)

(33)

| Nin | buyao | wei | wo | danxin. |
|-----|-------|------|-----|---------|
| you | don't | **PREP** | me | worry |

'Please don't worry about me.'
(ibid.)

In the +P/bare and +P/specific conditions, we used multiple-sluicing sentences where the first remnant, i.e., a *wh*-argument, was preceded by *shi* and the second remnant, i.e., a prepositional *wh*-remnant, was not. This implementation was based on the following considerations. First, *shi* is optional in front of prepositional *wh*-remnants (Wei 2004; Wang and Wu 2006; Park and Li 2013; Song 2016; Zhang and Overfelt 2019; Lee 2020). Second, our informal consultation with native speakers for sentences such as (34) and (35) revealed that *shi* with a prepositional *wh*-remnant was redundant, resulting in lower acceptability ratings.

(34) **Condition 3: +P/bare**

| Mei | ge | ren | dou | gei | mouwu | qi-guo-ming, | wo | zhishi | bu |
|-----|-----|--------|-----|------|--------|--------------|-----|--------|-----|
| every | CLF | person | all | PREP | something | name-PFV | I | just | not |

| zhidao | shi | shei | gei | shenme. |
|--------|-----|------|------|---------|
| know | **SHI** | who | PREP | what |

'Everyone named something, I just don't know who what.'

(35) **Condition 4: +P/specific**

| Mei | ge | nvhai | dou | gei | mou | ge | wanju | qi-guo-ming, |
|-----|-----|-------|-----|------|------|-----|-------|--------------|
| every | CLF | girl | all | PREP | some | CLF | toy | name-PFV |

| wo | zhishi | bu | zhidao | shi | na | ge | nvhai | gei | na | ge | wangju. |
|-----|--------|-----|--------|-----|-----|-----|-------|------|------|-----|---------|
| I | just | not | know | **SHI** | which | CLF | girl | PREP | which | CLF | toy |

'Every girl named some toy, I just don't know which girl which toy.'

In short, in the -P/specific, +P/bare, and +P/specific conditions, only the first *wh*-remnant was preceded by *shi*. In the -P/bare condition, on the other hand, both *wh*-remnants were accompanied by *shi*. This implementation of the distributions of *shi* in the test items was decided based on the previous literature, informal exploratory tests, and consultation with native speakers to eliminate or lessen the influence of *shi* on the

experimental results. We return to discussions on the distributions of *shi* in Experiment 2, presented in Section 3.2.

The distribution of test items followed a Latin square design, where four lists were created, and all items and fillers were randomized within each trial. Every participant saw a total of six items in each condition, thus 24 critical items in total. Additionally, 72 fillers were included in every list. Fifteen of those fillers served as control filler items, which included five degrees of acceptability from most natural to least natural. Each degree featured three sentences.[9] The purpose of including control fillers was to check whether participants used the rating scale correctly. Another 15 fillers were multiple *wh*-questions containing *wh*-arguments, *wh*-adjuncts, and prepositional *wh*-phrases. The remaining 42 fillers included various sentence constructions cited from the Modern Chinese Corpus compiled by the Center for Chinese Linguistics of Peking University (CCL Corpus) (Zhan et al. 2003). Accordingly, each participant rated a total of 96 experimental tokens.

Participants and Procedure

An acceptability judgment test was created using *PsychoPy 3* software (Peirce et al. 2019). Forty self-reported native speakers of MC (mean age = 25.5, SD = 1.89) studying at Tohoku University (Japan) were recruited via different social media channels. The task was deployed in lab, and thus participation occurred in person. Participants were instructed to read carefully and to rate the naturalness of sentences on a 7-point scale, from 1 (very unnatural) to 7 (very natural), based on their intuition. Additionally, they were informed that there were no "right" or "wrong" answers and that they should just follow their intuition. Each participant received a JPY 1000 Amazon gift card as compensation for their participation in this study, which lasted approximately 20 min. Based on the judgments participants gave to the control filler items, four participants were excluded for misusing the rating scale. Consequently, the data of 36 participants entered the statistical analysis. Lastly, a practice round with five sentences was conducted before participants began the critical trial. They were allowed to ask clarification questions about the procedure during this practice trial.

Predictions

For this experiment, we made the following predictions:

(36)    Prediction regarding PREPOSITIONHOOD
Multiple sluicing in which the second remnant is a prepositional *wh*-remnant should be rated significantly more acceptable than that in which the second remnant is a *wh*-argument.

(37)    Prediction regarding SPECIFICITY
Multiple sluicing in which the remnants are specific *wh*-phrases should be rated significantly more acceptable than that in which the remnants are bare *wh*-phrases.

The prediction in (36) is motivated by the cue-based retrieval approach to ellipsis and experimental results and discussions on multiple sluicing in English and Spanish (Cortés Rodríguez 2021, 2023). Like English and Spanish, MC lacks case morphology (Barrie and Li 2015), and multiple sluicing in this language should, therefore, rely on cues provided by prepositions so that the thematic roles can be properly discerned. Furthermore, constructions including a preposition consist of one more cue than those without a preposition. Consequently, the former cases are predicted to be more acceptable than the latter cases. Next, the prediction in (37) is motivated by the cue-based retrieval approach and the discussions in Bhattacharya and Simpson (2012) (see also Harris 2015, 2019). In MC, specific *wh*-arguments, such as *nage daxuesheng* 'which college student' and *nage xiangmu* 'which project' in (31), provide a set of cues for identifying the correlate–remnant pairing relation efficiently. By comparison, bare *wh*-arguments, such as *shei* 'who' and *shenme* 'what' in (30), lack the information provided by nominal restrictors, which are considered as cues in the previous literature. Without case morphology, the information provided by nominal restrictors should facilitate the processing of multiple-sluicing sentences, resulting in higher acceptability ratings.

3.1.2. Data Analysis and Results

The data were analyzed in statistics software R, Version 4.1.2 (R Core Team 2021). We employed an ordinal logistic regression, and in particular, we used the *clmm* function of the *ordinal* package (Christensen 2019). To find the model with the best fit, we implemented a manual backward model section process using the *anova* function. We started checking the full model, namely the one including all experimental factors and interactions as fixed effects, as well as random effects for both items and subjects with their maximal random slopes and respective interactions.[10] Then the model was progressively checked against a minimally simplified model until the model with the most complex random effect structure that would converge was reached (Barr et al. 2013). Here, we report the model with the best maximal fixed and random effect structure supported by the experimental data. The corresponding formula is provided in the tables with the statistical analysis.

Figure 1 shows the mean acceptability ratings obtained for the four experimental conditions; the results of its statistical analysis are presented in Table 1. Additionally, Table 2 provides the means and standard deviation of the individual conditions. The model yielded two main effects for PREPOSITIONHOOD and SPECIFICITY. Concerning PREPOSITIONHOOD, multiple sluicing sentences where a preposition was present in the non-initial *wh*-remnant were rated as significantly more acceptable. As for the main effect observed for SPECIFICITY, participants rated *specific* conditions as significantly more acceptable than *bare* ones. The direction of those two main effects and the lack of individual difference between Conditions 2 and 3 are indicative of an additive effect. There was no significant interaction between the factors. Finally, the overall mean ratings for multiple sluicing in MC were in the 3.7 range on a 7-point Likert, as shown in Figure 1. This result indicates that multiple sluicing is a marked construction in MC, just like that in English and Spanish (Cortés Rodríguez 2021, 2023).

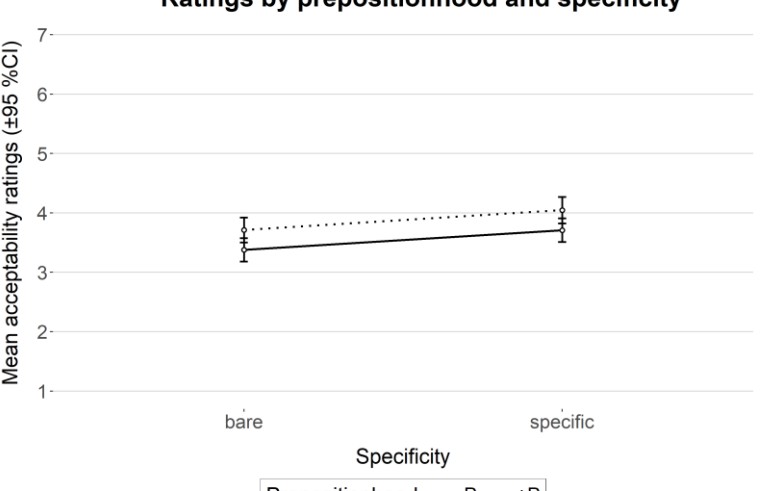

**Figure 1.** Mean acceptability ratings (n = 36). Error bars show 95% confidence interval.

**Table 1.** Cumulative Link Mixed Model fitted with the Laplace approximation.

|  | Estimate | Std. Error | z Value | Pr(>\|z\|) |  |
| --- | --- | --- | --- | --- | --- |
| PREPOSITIONHOOD (+*P*) | 0.6597 | 0.2527 | 2.611 | 0.00903 | ** |
| SPECIFICITY (*specific*) | 0.6265 | 0.1282 | 4.888 | $1.02 \times 10^{-6}$ | *** |

**Formula:** *rating ~ prepositionhood + specificity + (prepositionhood | subject) + (1 | item). item).* The significance levels used in across all experiments reported here are the following: $p < 0.05$ = *; $p < 0.01$ = **; $p < 0.001$ = ***.

**Table 2.** Mean acceptability ratings in Experiment 1.

| | | Condition | Rating (SD) |
|---|---|---|---|
| 1 | bare *wh* | **-P** bare *wh* | 3.36 (1.46) |
| 2 | specific *wh* | **-P** specific *wh* | 3.69 (1.47) |
| 3 | bare *wh* | **+P** bare *wh* | 3.70 (1.56) |
| 4 | specific *wh* | **+P** specific *wh* | 4.00 (1.65) |

The predictions made for this experiment are borne out. We will further discuss the results in Section 4.1.[11] In particular, we will discuss how the cue-retrieval approach suggested for parallel studies in English and German (Cortés Rodríguez 2023) can capture the differences observed here.

### 3.2. Experiment 2: The Distribution of shi

We conducted a series of experiments to examine the distribution of *shi* in multiple sluicing in MC, motivated by the following two points. On the one hand, since the previous literature reports varied distributions of *shi* in multiple sluicing, we were motivated to define the distributions based on the results of controlled experimentation. On the other hand, given that our implementation of the distribution of *shi* in Experiment 1 was determined based on a small set of informally collected judgments (besides our own intuition), we sought to confirm that our implementation could be supported by formal experimentation.

#### 3.2.1. Methods
Design and Materials

We conducted four sub-experiments in this series, all of which followed a 2 × 2 within-item and within-subject design. The two independent variables were (i) SHI-WH1, representing the presence or absence of *shi* accompanying the first *wh*-remnant, thus the factor levels were simply *yes* (*shi* is present) and *no* (*shi* is not present), and (ii) SHI-WH2, likewise modulating the presence or absence of *shi* in the second *wh*-remnants. The test conditions are illustrated in Table 3.

**Table 3.** The distribution of *shi* in each test condition.

| Condition | The Distribution of *shi* | |
|---|---|---|
| | SHI-WH1 | SHI-WH2 |
| 1 | yes | yes |
| 2 | yes | no |
| 3 | no | yes |
| 4 | no | no |

The four tested conditions in Experiment 1, i.e., -P/bare, -P/specific, +P/bare, and +P/specific, were separately tested in sub-experiments 1–4, respectively. Each of the sub-experiments tested all the possible distributions of *shi*, as presented in Table 3. Moreover, the sentence structure used across all experimental items in the four sub-experiments mirrored the same pattern introduced for Experiment 1. See (38)–(43) for example test items in each sub-experiment.

In sub-experiment 1, the distributions of *shi* in the -P/bare condition were tested, as shown in the example test item (38). In sub-experiment 2, the distributions of *shi* in the -P/specific condition were tested, as in (39). Sub-experiments 1 and 2 each included 24 critical items in four conditions.

(38)    **Sub-experiment 1 [nominal bare—nominal bare]**

| Mei | ge | ren | dou | wancheng-le | moushi, |
|---|---|---|---|---|---|
| every | CLF | person | all | complete-PFV | something |

'Everyone completed something,'

| | wo | zhishi | bu | zhidao | shi | shei | shi | shenme. |
|---|---|---|---|---|---|---|---|---|
| C1. | I | just | not | know | SHI | who | SHI | what |
| C2. | wo | zhishi | bu | zhidao | shi | shei | shenme. | |
| | I | just | not | know | SHI | who | what | |
| C3. | wo | zhishi | bu | zhidao | shei | shi | shenme. | |
| | I | just | not | know | who | SHI | what | |
| C4. | wo | zhishi | bu | zhidao | shei | shenme. | | |
| | I | just | not | know | who | what | | |

'I just don't know who what.'

(39) **Sub-experiment 2 [nominal specific—nominal specific]**

| Mei | ge | daxuesheng | dou | wancheng-le | yi | ge | xiangmu, |
|---|---|---|---|---|---|---|---|
| every | CLF | college.student | all | complete-PFV | one | CLF | project |

'Every college student completed a project,'

| | wo | zhishi | bu | zhidao | shi | na | ge | daxuesheng | shi | na | ge | xiangmu. |
|---|---|---|---|---|---|---|---|---|---|---|---|---|
| C1. | I | just | not | know | SHI | which | CLF | college.student | SHI | which | CLF | project |
| C2. | wo | zhishi | bu | zhidao | shi | na | ge | daxuesheng | na | ge | xiangmu. | |
| | I | just | not | know | SHI | which | CLF | college.student | which | CLF | project | |
| C3. | wo | zhishi | bu | zhidao | na | ge | daxuesheng | shi | na | ge | xiangmu. | |
| | I | just | not | know | which | CLF | college.student | SHI | which | CLF | project | |
| C4. | wo | zhishi | bu | zhidao | na | ge | daxuesheng | na | ge | xiangmu. | |
| | I | just | not | know | which | CLF | college.student | which | CLF | project | |

'I just don't know which college student which project.'

In sub-experiment 3, we tested the distributions of *shi* in the +P/bare condition, as exemplified in (40). Additionally, this sub-experiment tested the distributions of *shi* in cases where the second correlate was an adjunct, as in (41). We decided to include cases with adjuncts for the following reasons. First, although cases with adjuncts have been discussed in the previous literature, they have not been experimentally tested. Second, the adjuncts used in multiple sluicing in MC all have an underlying prepositional structure. For instance, the *wh*-adjunct in (41) *zai heshi* 'at when' includes the preposition *zai*. Accordingly, sub-experiment 3 had 32 critical items: 16 including a PP argument and another 16 including an adjunct. Similarly, sub-experiment 4 also contained 32 item quadruplets: 16 including a PP argument in the +P/specific condition and another 16 including a specific adjunct as the second *wh*-remnant, as exemplified in (42) and (43), respectively. The sub-experiments containing adjuncts are referred to as sub-experiments 3' and 4', respectively, both including an equal number of locative and temporal adjuncts (i.e., 8 locative and 8 temporal).

(40)    **Sub-experiment 3 [nominal bare—prepositional bare]**

| Mei | ge | ren | dou | gei | mouwu | qi-guo-ming, |
|---|---|---|---|---|---|---|
| every | CLF | person | all | PREP | something | name-PFV |

'Everyone named something,'

| | wo | zhishi | bu | zhidao | shi | shei | shi | gei | shenme. |
|---|---|---|---|---|---|---|---|---|---|
| C1. | I | just | not | know | SHI | who | SHI | PREP | what |
| C2. | wo | zhishi | bu | zhidao | shi | shei | gei | shenme. | |
| | I | just | not | know | SHI | who | PREP | what | |
| C3. | wo | zhishi | bu | zhidao | shei | shi | gei | shenme. | |
| | I | just | not | know | who | SHI | PREP | what | |
| C4. | wo | zhishi | bu | zhidao | shei | gei | shenme. | | |
| | I | just | not | know | who | PREP | what | | |

'I just don't know who what.'

(41)  **Sub-experiment 3' [nominal bare—bare adjunct]**

| | Mei | ge | ren | dou | zai | moushi | qu-guo | Beijing, |
|---|---|---|---|---|---|---|---|---|
| | every | CLF | person | all | at | sometime | go-PFV | Beijing |

'Everyone went to Beijing at sometime,'

| | | | | | | | | |
|---|---|---|---|---|---|---|---|---|
| C1. | wo | zhishi | bu | zhidao | shi | shei | shi | zai | heshi. |
| | I | just | not | know | **SHI** | who | **SHI** | at | when |
| C2. | wo | zhishi | bu | zhidao | shi | shei | zai | heshi. |
| | I | just | not | know | **SHI** | who | at | when |
| C3. | wo | zhishi | bu | zhidao | shei | shi | zai | heshi. |
| | I | just | not | know | who | **SHI** | at | when |
| C4. | wo | zhishi | bu | zhidao | shei | zai | heshi. |
| | I | just | not | know | who | at | when |

'I just don't know who when.'

(42)  **Sub-experiment 4 [nominal specific—prepositional specific]**

| Mei | ge | nvhai | dou | gei | mou | ge | wanju | qi-guo-ming, |
|---|---|---|---|---|---|---|---|---|
| every | CLF | girl | all | PREP | some | CLF | toy | name-PFV |

'Every girl named some toy,'

| | | | | | | | | | | | | |
|---|---|---|---|---|---|---|---|---|---|---|---|---|
| C1. | wo | zhishi | bu | zhidao | shi | na | ge | nvhai | shi | gei | na | ge | wangju. |
| | I | just | not | know | **SHI** | which | CLF | girl | **SHI** | PREP | which | CLF | toy |
| C2. | wo | zhishi | bu | zhidao | shi | na | ge | nvhai | gei | na | ge | wangju. |
| | I | just | not | know | **SHI** | which | CLF | girl | PREP | which | CLF | toy |
| C3. | wo | zhishi | bu | zhidao | na | ge | nvhai | shi | gei | na | ge | wangju. |
| | I | just | not | know | which | CLF | girl | **SHI** | PREP | which | CLF | toy |
| C4. | wo | zhishi | bu | zhidao | na | ge | nvhai | gei | na | ge | wanju. |
| | I | just | not | know | which | CLF | girl | PREP | which | CLF | toy |

'I just don't know which girl which toy.'

(43)  **Sub-experiment 4' [nominal specific—specific adjunct]**

| Mei | ge | xuesheng | dou | zai | mou | ge | shijian | qu-guo | Beijing, |
|---|---|---|---|---|---|---|---|---|---|
| every | CLF | student | all | at | some | CLF | time | go-PFV | Beijing |

'Every student went to Beijing at some time,'

| | | | | | | | | | | | |
|---|---|---|---|---|---|---|---|---|---|---|---|
| C1. | wo | zhishi | bu | zhidao | shi | na | ge | xuesheng | shi | zai | shenme | shijian. |
| | I | just | not | know | **SHI** | which | CLF | student | **SHI** | at | what | time |
| C2. | wo | zhishi | bu | zhidao | shi | na | ge | xuesheng | zai | shenme | shijian. |
| | I | just | not | know | **SHI** | which | CLF | student | at | what | time |
| C3. | wo | zhishi | bu | zhidao | na | ge | xuesheng | shi | zai | shenme | shijian. |
| | I | just | not | know | which | CLF | student | **SHI** | at | what | time |
| C4. | wo | zhishi | bu | zhidao | na | ge | xuesheng | zai | shenme | shijian. |
| | I | just | not | know | which | CLF | student | at | what | time |

'I just don't know which student at what time.'

The distribution of test items followed a Latin square design, where four lists were created for each sub-experiment, and all items and fillers were randomized within each trial. In sub-experiments 1 and 2, every participant saw a total of six items in each condition, thus a total of 24 critical items. In sub-experiments 3 and 4, every participant saw a total of four items in each condition of prepositions and adjuncts, thus a total of 32 critical items. Additionally, 72 fillers were included in every list, resulting in a final amount of 96 experimental tokens that each participant rated in sub-experiments 1 and 2 and a final amount of 104 experimental tokens that each participant rated in sub-experiments 3 and 4.

Participants and Procedure

We conducted four web-based acceptability judgment experiments. As in Experiment 1, we used *PsychoPy 3* as the experiment creation software. For online participation, we hosted the experiments in *Pavlovia*.[12] Using WeChat, the following number of participants were recruited: 32 for sub-experiment 1, 31 for sub-experiment 2, 30 for sub-experiment 3, and 33 for sub-experiment 4. Participants were self-reported adult native speakers of Mandarin Chinese. They were not informed of the purpose of the experiment; their

instructions were only to rate the naturalness of the presented sentences on a 7-point scale from 1 (very unnatural) to 7 (very natural) based on their intuition. They were also informed that there was no "right" or "wrong" answer. Each participant received CNY 30 as cash remuneration for their participation in the study, which lasted approximately 20 min. Based on the judgments the participants gave to a set of control fillers, which were the same control fillers as in Experiment 1, the following number of participants were excluded from the analysis for misusing the scale: 5 from sub-experiment 1, 3 from sub-experiment 2, 5 from sub-experiment 3, and 8 from sub-experiment 4. Consequently, the data from 105 participants (27 from sub-experiment 1; 28 from sub-experiment 2; 25 from sub-experiment 3; 25 from sub-experiment 4) were included in the analysis. Lastly, a practice round with five sentences was conducted before participants began the critical trial. During this practice trial, they were allowed to ask clarification questions about the procedure.

Predictions Regarding the Distribution of *shi*

In this experiment, we made the following predictions:

(44) a. The presence or absence of *shi* should significantly influence the acceptability of multiple sluicing sentences with bare *wh*-arguments.

    b. The presence or absence of *shi* should not significantly influence the acceptability of multiple sluicing sentences with specific *wh*-arguments, prepositional *wh*-arguments, or *wh*-adjuncts.

Since the influence of the different distributions of *shi* on the acceptability of multiple sluicing in MC has not been comprehensively discussed in the previous literature, the predictions in (44) are motivated by the reported usage of *shi* in single sluicing in the language, as reviewed in Section 2.1.

### 3.2.2. Data Analysis and Results

The data of the four sub-experiments were analyzed using the same procedures as in Experiment 1. First, we present the descriptive statistics for sub-experiments 1 and 2. Figures 2 and 3 show the mean acceptability ratings (±95% CI) obtained for the four experimental conditions, and the results of its statistical analysis are presented in Tables 4 and 5, respectively. Additionally, Tables 6 and 7 provide the means and standard deviation of the individual conditions.

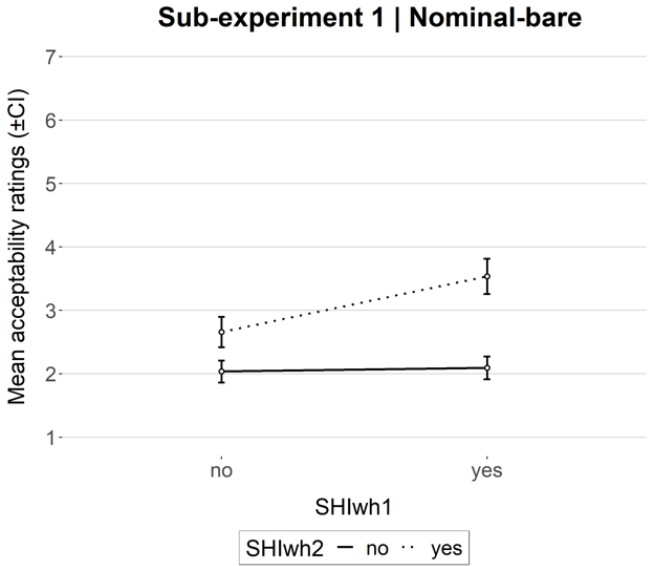

**Figure 2.** Mean acceptability ratings (n = 27). Error bars show 95% confidence interval.

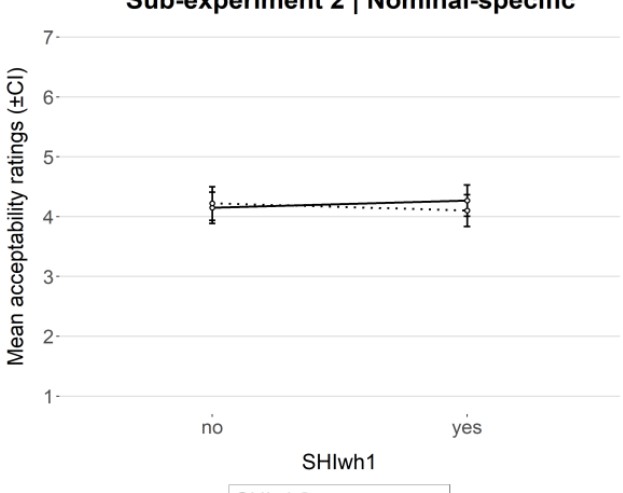

**Figure 3.** Mean acceptability rating (n = 28). Error bars show 95% confidence interval.

On the one hand, the model for the data in sub-experiment 1 yielded a main effect for SHI-WH2, as well as an interaction. No significant effect was observed for SHI-WH1. Concerning SHI-WH2, the results showed that multiple sluicing configurations where a *shi* was present in the non-initial *wh*-remnant produced significantly more acceptable sentences. Given the significant interaction, a post hoc *Tukey* test was performed to check for individual comparison between each level. Crucially, the only comparison levels that did not show a significant difference were the conditions where the non-initial *wh*-remnant was not accompanied by *shi*.

On the other hand, the results for sub-experiment 2 did not show any significant effect.

**Table 4.** Cumulative link mixed model fitted with the Laplace approximation. (Sub-experiment 1 | Nominal-bare.)

|  | Estimate | Std. Error | z Value | Pr (>|z|) | |
|---|---|---|---|---|---|
| SHI-WH1 (*yes*) | 0.1664 | 0.2810 | 0.592 | 0.553793 | |
| SHI-WH2 (*yes*) | 1.0798 | 0.2966 | 3.641 | 0.000272 | *** |
| SHI-WH1: SHI-WH2 | 1.3254 | 0.3103 | 4.271 | $1.94 \times 10^{-5}$ | *** |

**Formula**: *rating ~ SHIwh1*SHIwh2 + (SHIwh1+SHIwh2 | subject) + (1 | item).*

**Table 5.** Cumulative link mixed model fitted with the Laplace approximation. (Sub-experiment 2 | Nominal-specific.)

|  | Estimate | Std. Error | z Value | Pr (>|z|) |
|---|---|---|---|---|
| SHI-WH1 (*yes*) | 0.05954 | 0.22163 | 0.269 | 0.788 |
| SHI-WH2 (*yes*) | −0.18676 | 0.22988 | −0.812 | 0.417 |

**Formula**: *rating ~ SHIwh1+SHIwh2 + (SHIwh1*SHIwh2 | subject) + (1 | item)*

**Table 6.** Mean acceptability ratings in sub-experiment 1.

|  | Condition | | Rating (SD) |
|---|---|---|---|
| 1 | **shi** bare *wh* | **shi** bare *wh* | 3.54 (1.78) |
| 2 | **shi** bare *wh* | bare *wh* | 2.09 (1.16) |
| 3 | bare *wh* | **shi** bare *wh* | 2.66 (1.55) |
| 4 | bare *wh* | bare *wh* | 2.04 (1.12) |

**Table 7.** Mean acceptability ratings in sub-experiment 2.

|   | Condition | | Rating (SD) |
|---|---|---|---|
| 1 | **shi** specific *wh* | **shi** specific *wh* | 4.10 (1.74) |
| 2 | **shi** specific *wh* | specific *wh* | 4.27 (1.72) |
| 3 | specific *wh* | **shi** specific *wh* | 4.22 (1.85) |
| 4 | specific *wh* | specific *wh* | 4.15 (1.70) |

Second, Figure 4 illustrates the mean acceptability ratings (±95% CI) obtained for sub-experiments 3 and 3′. Two separate models were calculated for the items containing a prepositional phrase as the non-initial *wh*-remnant and for the items where the non-initial *wh*-remnant was an adjunct (a locative or temporal adverb, to be precise). Tables 8 and 9 demonstrate the statistical analysis for sub-experiments 3 and 3′, respectively. Furthermore, the means and standard deviation of the individual conditions are presented in Tables 10 and 11.

In both cases, the results showed a main effect SHI-WH1; that is, significantly higher acceptability ratings were obtained for conditions where the initial *wh*-phrase, i.e., a bare *wh*-argument, was preceded by *shi*.

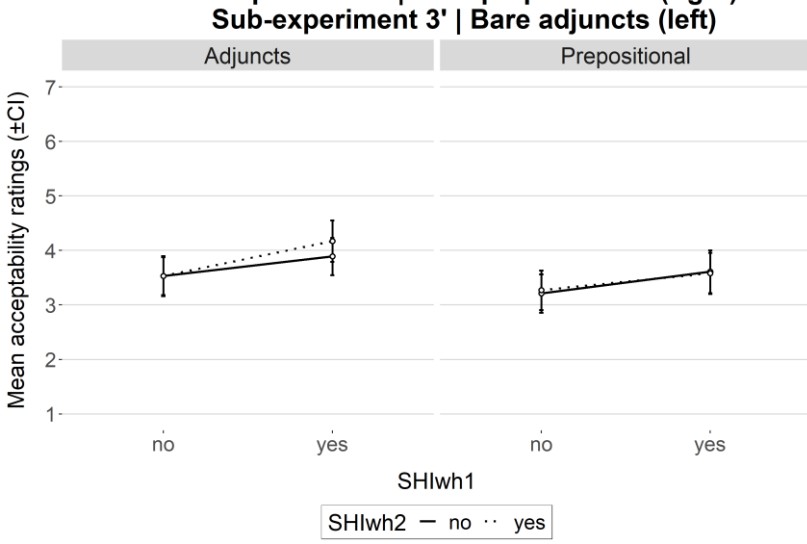

**Figure 4.** Mean acceptability ratings (n = 25). Error bars show 95% confidence interval.

**Table 8.** Cumulative link mixed model fitted with the Laplace approximation. (Sub-experiment 3 | Bare-Prepositional.)

|   | Estimate | Std. Error | z Value | Pr (>\|z\|) | |
|---|---|---|---|---|---|
| SHI-WH1(*yes*) | 0.4840 | 0.2379 | 2.034 | 0.042 | * |

**Formula**: rating ~ SHIwh1 + (SHIwh1 | subject) + (1 | item)

**Table 9.** Cumulative link mixed model fitted with the Laplace approximation. (Sub-experiment 3′ | Bare-Adjunct.)

|   | Estimate | Std. Error | z Value | Pr (>\|z\|) | |
|---|---|---|---|---|---|
| SHI-WH1(*yes*) | 0.6939 | 0.2766 | 2.509 | 0.0121 | * |

**Formula**: rating ~ SHIwh1 + (SHIwh1 | subject) + (1 | item)

**Table 10.** Mean acceptability ratings in sub-experiment 3.

| | Condition | | Rating (SD) |
|---|---|---|---|
| 1 | **shi** bare *wh* | **shi** bare prepositional *wh* | 3.73 (1.91) |
| 2 | **shi** bare *wh* | bare prepositional *wh* | 3.66 (1.98) |
| 3 | bare *wh* | **shi** bare prepositional *wh* | 3.37 (1.82) |
| 4 | bare *wh* | bare prepositional *wh* | 3.32 (1.76) |

**Table 11.** Mean acceptability ratings in sub-experiment 3'.

| | Condition | | Rating (SD) |
|---|---|---|---|
| 1 | **shi** bare *wh* | **shi** bare adjunct *wh* | 4.26 (1.87) |
| 2 | **shi** bare *wh* | bare adjunct *wh* | 3.94 (1.73) |
| 3 | bare *wh* | **shi** bare adjunct *wh* | 3.75 (1.73) |
| 4 | bare *wh* | bare adjunct *wh* | 3.67 (1.82) |

Third, the mean acceptability ratings obtained for sub-experiments 4 and 4' are presented in Figure 5. The statistical analysis for the model containing specific prepositional phrases is given in Table 12; the means and standard deviation of the individual conditions are provided in Table 13. The results for the model including specific adjunct phrases are provided in Table 14; the means and standard deviation are presented in Table 15. Similar to the results obtained in sub-experiment 2, none of the factors reached significance in sub-experiments 4 and 4'.

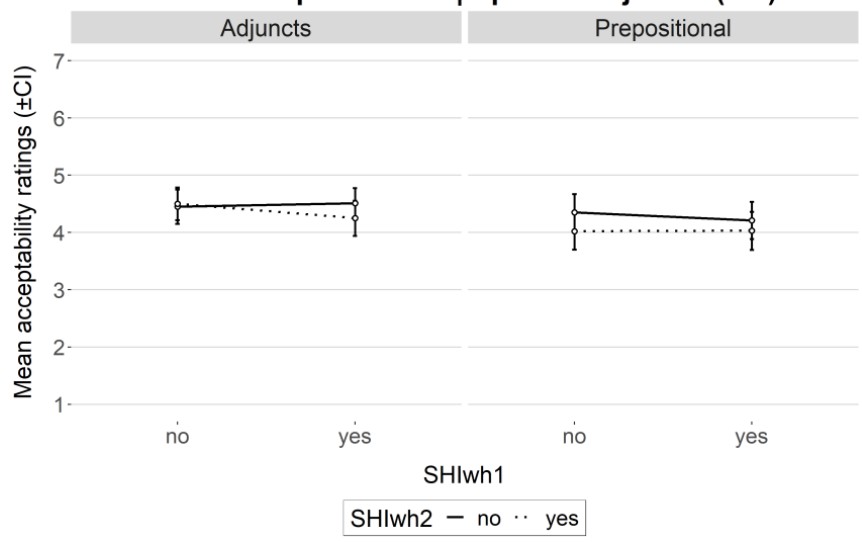

**Figure 5.** Mean acceptability ratings (n = 25). Error bars show 95% confidence interval.

**Table 12.** Cumulative link mixed model fitted with the Laplace approximation. (Sub-experiment 4 | Specific-Prepositional.)

| | Estimate | Std. Error | z Value | Pr (>|z|) |
|---|---|---|---|---|
| SHI-WH1 (*yes*) | −0.3233 | 0.1820 | −1.776 | 0.0757 |

**Formula**: rating ~ SHIwh1 + (SHIwh1 | subject) + (1 | item)

**Table 13.** Mean acceptability ratings in sub-experiment 4.

| | Condition | | Rating (SD) |
|---|---|---|---|
| 1 | **shi** specific *wh* | **shi** specific prepositional *wh* | 4.03 (1.67) |
| 2 | **shi** specific *wh* | specific prepositional *wh* | 4.21 (1.64) |
| 3 | specific *wh* | **shi** specific prepositional *wh* | 4.02 (1.61) |
| 4 | specific *wh* | specific prepositional *wh* | 4.35 (1.62) |

**Table 14.** Cumulative link mixed model fitted with the Laplace approximation. (Sub-experiment 4′ | Specific-Adjunct.)

| | Estimate | Std. Error | z Value | Pr (>\|z\|) |
|---|---|---|---|---|
| SHI-WH1 (*yes*) | −0.1568 | 0.2284 | −0.687 | 0.492 |

**Formula**: rating ~ SHIwh2 + (SHIwh1+ SHIwh2 | subject) + (1 | item)

**Table 15.** Mean acceptability ratings in sub-experiment 4′.

| | Condition | | Rating (SD) |
|---|---|---|---|
| 1 | **shi** specific *wh* | **shi** adjunct specific *wh* | 4.25 (1.55) |
| 2 | **shi** specific *wh* | adjunct specific *wh* | 4.51 (1.35) |
| 3 | specific *wh* | **shi** adjunct specific *wh* | 4.50 (1.43) |
| 4 | specific *wh* | adjunct specific *wh* | 4.45 (1.51) |

We will further discuss the results of this series of experiments in Section 4.2.

## 4. General Discussion

### 4.1. On Experiment 1

As presented in Section 3.1, the results of Experiment 1 show that prepositionhood and specificity improve the acceptability of multiple sluicing in MC, which aligns with the cue-retrieval approach to ellipsis. First, the presence of a prepositional *wh*-remnant as the second remnant makes multiple sluicing sentences more acceptable, which can be attributed to the cues provided by prepositions in discerning argument structures. Prepositions in MC facilitate the processing of multiple sluicing sentences, resulting in higher acceptability ratings, which is in accordance with the observations made in Cortés Rodríguez (2021, 2023) for multiple sluicing in English and Spanish. These three languages all lack case morphology, making cues contributed by prepositions necessary in processing multiple sluicing constructions.

The experimental results also demonstrate that, in MC, multiple sluicing sentences with specific *wh*-arguments are significantly more acceptable than those with bare *wh*-arguments, which can be attributed to the additional cues supplied by specific *wh*-phrases. Specific *wh*-phrases composed of *which* and an NP restrictor are discourse linked, presupposing that there is a set of individuals and objects salient to discourse participants (Pesetsky 1987; Comorovski 1996). The experiment contains cases of multiple sluicing with a universal quantifier and an existential quantifier as correlates, which are more complex than multiple sluicing sentences with two existential quantifiers because the former forces pair-list interpretation, while the latter produces single-pair reading. Processing the former cases is difficult in formal experimental settings where no contexts are provided. Discourse-linked *wh*-phrases contribute to the association of multiple sluicing sentences with concrete contexts and discourse containing salient sets of individuals and objects, which can facilitate processing. For example, *nage daxuesheng* 'which college student' and *nage xiangmu* 'which project' in (31) presuppose that there is a set of college students and projects in the discourse. Moreover, nominal restrictors help the establishment of the correlate–remnant

matching relation and the retrieval of information from antecedent clauses. On the other hand, bare *wh*-arguments cannot refer to any pre-established information in the discourse, thus increasing the difficulty in processing the relevant multiple sluicing sentences. For instance, *shei* 'who' and *shenme* 'what' in (30) refer to all humans and things.

Furthermore, although our experiment shows that specific *wh*-phrases improve the acceptability of multiple sluicing in MC, the specificity effect is not observed in English or Spanish (Cortés Rodríguez 2021, 2023). Since our experiment is modeled after Cortés Rodríguez (2021, 2023), we would like to make a preliminary assumption explaining this difference. We conjecture that cues function differently in different languages in accordance with language-specific properties. In a language with rich case morphology, cues provided by overt case marking are sufficient to facilitate the processing of elliptical constructions. For instance, in German, the overall mean ratings for multiple sluicing constructions are in the 5 range on a 7-point Likert scale, irrespective of the presence or absence of a preposition or a specific *wh*-argument (Cortés Rodríguez 2023). The fact that the preposition and specificity effects are not observed in German multiple sluicing indicates that cues provided by case markers are sufficient. On the other hand, in languages with poor case morphology such as English, Spanish, and MC, syntactic cues provided by a preposition are effective in facilitating the processing of multiple sluicing constructions, as discussed in Cortés Rodríguez (2021, 2023) and the present paper. Furthermore, discourse-related cues provided by discourse-linked *wh*-phrases work differently in different languages. In MC, a discourse-oriented language (Huang 1984; Shei 2019), discourse-related cues could play a significant role in ellipsis processing.[13] English and Spanish, on the other hand, are sentence-oriented languages (Wakabayashi 2002), where discourse-related cues may not significantly affect the processing of multiple sluicing in the two languages.[14]

In summary, the results of our experiment on MC support the cue-retrieval analysis explaining the differences in acceptability ratings of cross-linguistic multiple sluicing constructions (Cortés Rodríguez 2021, 2023).

*4.2. On Experiment 2*

This section discusses the results of the series of sub-experiments we conducted in Section 3.2. In sub-experiment 1 with bare *wh*-arguments, the overall result was that the presence of *shi* affected acceptability ratings. First and foremost, condition 1 with *shi* preceding each bare *wh*-argument received the highest rating, supporting our implementation of the distribution of *shi* in Experiment 1. Moreover, in conditions 2–4, where one of the bare *wh*-arguments or neither were preceded by *shi*, the ratings were rather low, demonstrating that the relevant constructions were completely unacceptable.[15] In accordance with the results from sub-experiment 1, we maintain that bare *wh*-arguments in multiple sluicing in MC require the support of *shi*, which is in line with the observations made in the previous literature.

In sub-experiment 2 with specific *wh*-arguments, the general result was that there were only minimal differences among the different distributions of *shi*. Nevertheless, as demonstrated in Table 7, the mean acceptability rating for condition 2 was the highest among the four conditions, supporting our implementation of the distribution of *shi* in Experiment 1. Interestingly, condition 1 with *shi* accompanying each specific *wh*-argument received the lowest rating, which was in direct contrast to the result from sub-experiment 1 with bare *wh*-arguments. Based on the results from sub-experiment 2, we claim that specific *wh*-arguments do not necessarily require the support of *shi*. Moreover, these results confirm the observations made for single sluicing in MC, namely, the occurrence of *shi* is optional in front of specific *wh*-arguments. The results from sub-experiments 1 and 2 further consolidate our conclusion from Experiment 1 that specificity is an ameliorating factor in multiple sluicing in MC, since the overall mean in sub-experiment 2 was higher than that in sub-experiment 1.

In sub-experiments 3 and 3' including bare prepositional *wh*-arguments and bare *wh*-adjuncts, the results showed that acceptability ratings were significantly higher when the

first remnant, i.e., the bare *wh*-argument, was preceded by *shi*. This result is not surprising because a bare *wh*-argument requires *shi*-support. Furthermore, the presence or absence of the second *shi* did not significantly affect the acceptability ratings, indicating that the distribution of *shi* we used in Experiment 1 was correct. Based on the results, we claim that the occurrence of *shi* is optional with prepositional *wh*-phrases and *wh*-adjuncts in multiple sluicing, which parallels the observations made for single sluicing in MC. Moreover, the results strengthen the argument that bare *wh*-arguments require the support of *shi*.

In sub-experiments 4 and 4′, which included specific prepositional *wh*-arguments and specific *wh*-adjuncts, the overall result showed that there were minimal differences among the different distributions of *shi*, just as in the results of sub-experiment 2 with specific *wh*-arguments. Thus, we claim that specific *wh*-remnants do not require the support of *shi*.

In general, the influence of the distributions of *shi* on the acceptability of multiple sluicing in MC is related to the nature of the *wh*-remnants; that is, the presence of *shi* in front of bare *wh*-arguments significantly improved acceptability ratings, whereas *shi* in front of specific *wh*-arguments, prepositional *wh*-phrases, and *wh*-adjuncts did not influence acceptability ratings significantly. In other words, only bare *wh*-arguments obligatorily require *shi*-support.[16] With regard to the obligatory or optional presence of *shi* in front of *wh*-remnants in sluicing and multiple sluicing in MC, the previous literature has provided various explanations, all of which are related to different theoretical analyses of sluicing constructions, i.e., the pseudo-sluicing analysis and the *wh*-movement followed by TP-ellipsis analysis (see the referenced literature in Section 2.1 for details). It is beyond the scope of this paper to provide a definite answer to this line of inquiry. Nevertheless, the present study lays a solid empirical ground for further discussion.

Before concluding this paper, we would like to mention that the results of our experiments do not seem to favor the pseudo-sluicing analysis, which involves the conjunction of two copular clauses with null subjects, as discussed in Adams and Tomioka (2012). Consider (45) and (46):

(45)  | Mouren    | tou-le     | tade  | yi     | yang   | dongxi, | wo  | xiang | zhidao | [*pro* |
      | someone   | steal-PFV  | his   | one    | CLF    | thing   | I   | want  | know   | he     |
      | *(shi)    | shei]      | yiji  | [*pro* | *(shi) | shenme] |     |       |        |        |
      | be        | who        | and   | it     | be     | what    |     |       |        |        |

'Someone stole one of his belongings, and I wonder who he was and what it was'

(46)  | Laoshi    | chufa-le     | mouren,  | wo   | xiang | zhidao  | [*pro* | *(shi) | shei] |
      | teacher   | punish-PFV   | someone  | I    | want  | know    | he     | be     | who   |
      | yiji      | [*pro*       | (shi)    | wei  | shenme] |       |        |        |       |
      | and       | that         | be       | for  | what    |       |        |        |       |

'Teacher punished someone, and I wonder who he was and why that was'

Sentences in (45) and (46) illustrate the pseudo-sluicing analysis of example (3) with two *wh*-arguments and (17) with a *wh*-argument and a *wh*-adjunct, respectively. As predicted by the pseudo-sluicing analysis, the conjunction of two copular clauses is fully acceptable. That is, (45) and (46) are equally acceptable. Consequently, the pseudo-sluicing analysis can neither explain the degraded acceptability judgments nor capture the differences between *wh*-arguments and *wh*-adjuncts observed in our experiments. The detailed theoretical analysis of multiple sluicing in MC is left for future research.

## 5. Conclusions

Multiple sluicing constructions in MC have been investigated in terms of its general acceptability and distributions of *shi*. The previous literature has not conducted extensive examinations into either of these two aspects. Moreover, previous arguments were largely based on informal data collection. The current study advances the research on multiple sluicing in MC by initiating experimental studies, following Cortés Rodríguez (2021, 2023). The experiments presented in this paper show four important findings. First, similar to English and Spanish, multiple sluicing in MC was confirmed to be a marked construction

with acceptability ratings in the 3.7 range on a 7-point Likert scale. Second, factors like the presence of prepositions and specific *wh*-remnants were found to improve the overall acceptability of these constructions. Third, the distribution of *shi* was shown to have an effect on the acceptability of multiple sluicing. The presence of *shi* preceding bare *wh*-arguments significantly improved acceptability ratings, while *shi* in front of specific *wh*-arguments, prepositional *wh*-remnants, and adjunct *wh*-remnants did not significantly influence acceptability ratings. Finally, the (non-)optionality of *shi* in multiple sluicing parallels that found in single sluicing in MC. Based on these findings, we argue that the ameliorating effects of prepositionhood and specificity can be explained by a cue-retrieval approach to ellipsis. Specific *wh*-phrases, adjuncts, and prepositions provide cues to help the retrieval of information from antecedent clauses. Bare *wh*-arguments, on the other hand, increase the processing burden because of a lack of cues. Our experimental findings are in line with the cross-linguistic experimental studies of Cortés Rodríguez (2021, 2023). Multiple sluicing constructions are marked in languages with poor case morphology, such as English, Spanish, and MC. On the other hand, in languages with rich case morphology, such as German (e.g., Merchant 2006; Richards 2010; Cortés Rodríguez 2023) and Japanese (Takahashi 1994), multiple sluicing constructions are more acceptable. We further argue that the need for *shi*-support depends on the nature of *wh*-phrases; that is, only bare *wh*-arguments obligatorily require support from *shi*. This paper contributes to the study on multiple sluicing in MC by providing experimental evidence, thereby laying a solid foundation for further theoretical analyses.

**Author Contributions:** Conceptualization, X.B., Á.C.R. and D.T.; methodology, X.B. and Á.C.R.; software, Á.C.R.; data collection, X.B.; data analysis, Á.C.R.; writing—original draft preparation, X.B. and Á.C.R.; writing—review and editing, X.B., Á.C.R. and D.T.; revision, X.B., Á.C.R., D.T.; funding acquisition, X.B. All authors have read and agreed to the published version of the manuscript.

**Funding:** This work was supported by JST SPRING, Grant Number JPMJSP2114, and JST, the establishment of university fellowships towards the creation of science technology innovation, Grant Number JPMJFS2102.

**Institutional Review Board Statement:** The study was conducted in accordance with the Declaration of Helsinki, and approved by the Ethics Committee of Graduate School of International Cultural Studies of Tohoku University (date of approval: 20 December 2021).

**Informed Consent Statement:** Informed consent was obtained from all subjects involved in the study.

**Data Availability Statement:** The data are available upon request to the first author.

**Acknowledgments:** We would like to thank James Griffiths for commenting on a previous version of this article. We are very grateful to the two anonymous reviewers for their detailed comments and suggestions for improvement. We are also grateful for all the informants and participants who share their knowledge with us. Especially, we thank the following native speakers of MC for their comments and suggestions: Yue Wang, Huanhuan Feng, and Changwei Zhang. All remaining errors and shortcomings are necessarily our own. Finally, we acknowledge support from the Open Access Publication Fund of the University of Tübingen.

**Conflicts of Interest:** The authors declare that they have no competing interests.

## Abbreviations

CLF for classifier; COP for copula; FOC for focus; GEN for genitive; PFV for perfective; PREP for preposition; PRF for perfect; Q for question particle.

## Notes

[1] The symbols used in this paper to indicate the degree of degradation of multiple sluicing sentences are as follows: * > *? > ?? > ? > [no symbol] (from completely degraded to completely acceptable).

[2] Regarding the MC examples in this paper, some of the words have received different glossing in the previous literature. For instance, *yiyang* is glossed as *a* in Wang and Han (2018) but as *one*-CL in Adams and Tomioka (2012). For expository

reasons, we use a unified glossing system throughout this paper, following the general guidelines of the Leipzig Glossing Rules (http://www.eva.mpg.de/lingua/resources/glossing-rules.php, accessed on 12 January 2023). We thank an anonymous reviewer for reminding us of this.

3   The previous literature provides different explanations for the obligatory presence of *shi* in front of bare *wh*-arguments. For example, Wei (2004) discusses that *shi* is obligatory when the *wh*-remnants are non-predicative and optional when the *wh*-remnants are predicative. See the referenced literature for details.

4   An exploratory test was conducted to examine whether bare *wh*-arguments must be preceded by *shi*. Seven native speakers of MC judged the acceptability of the following sentences on a 7-point Likert scale (1 being 'completely unacceptable' and 7 being 'completely acceptable'). The average rating for (ia) with *shi* preceding each bare *wh*-argument is 4.29. The average rating for (ib) with *shi* preceding only the first bare *wh*-argument is 1.86.

(i)   

| Mouren | tou-le | Xiaoming | de | yi | yang | dongxi, |
|--------|--------|----------|-----|-----|------|---------|
| someone | steal-PFV | Xiaoming | GEN | one | CLF | thing |

'Someone stole a thing of Xiaoming's,'

| a. ?? | wo | xiang | zhidao | shi | shei | shi | shenme. |
|-------|-----|-------|--------|-----|------|-----|---------|
|       | I | want | know | SHI | who | SHI | what |

| b. * | wo | xiang | zhidao | shi | shei | shenme. |
|------|-----|-------|--------|-----|------|---------|
|      | I | want | know | SHI | who | what |

'lit. I want to know who what.'

5   Let us note that (16) and (17) actually involve sprouting, where the *wh*-adjuncts have implicit correlates in the corresponding antecedent clauses (see also Wei 2004; Takahashi and Lin 2012; Park and Li 2013; Wang 2018). See Chung et al. (1995) for a discussion on sprouting.

6   Another combination, i.e., that of two *wh*-adjuncts, is discussed and allowed in multiple sluicing in MC. Since this combination is not related to the present discussion, please see Wang (2018) and Wang and Han (2018) for details.

7   Ten native speakers of MC judged the acceptability of the following sentences on a 7-point Likert scale (1 being 'completely unacceptable' and 7 being 'completely acceptable'). The average rating of (ia) with *shi* preceding each *wh*-argument is 3.6. The average rating of (ib) with *shi* preceding the first *wh*-argument is 4.2.

(i)   Context: There are three boys: Zhangsan, Lisi, and Wangwu. Each boy bought a different kind of fruit yesterday. The speaker is aware of this context.

| Mei | ge | nanhai | dou | mai-le | yi | zhong | shuiguo, |
|-----|-----|--------|-----|--------|-----|-------|----------|
| every | CLF | boy | all | buy-PFV | one | CLF | fruit |

'Every boy bought one kind of fruit,'

| a. | wo | xiang | zhidao | shi | na | ge | nanhai | shi | na | zhong | shuiguo. |
|----|-----|-------|--------|-----|-----|-----|--------|-----|-----|-------|----------|
|    | I | want | know | SHI | which | CLF | boy | SHI | which | CLF | fruit |

| b. | wo | xiang | zhidao | shi | na | ge | nanhai | na | zhong | shuiguo. |
|----|-----|-------|--------|-----|-----|-----|--------|-----|-------|----------|
|    | I | want | know | SHI | which | CLF | boy | which | CLF | fruit |

'I want to know which boy which kind of fruit.'

8   In MC, a prepositional phrase follows a verb when it indicates the location of the subject as a result of the action (Li and Thompson 1981; Ross and Ma 2014), as in (i).

(i)   

| Ta | tiao | dao | chuang | shang. |
|----|------|-----|--------|--------|
| he | jump | PREP | bed | on |

'He jumped onto the bed.'
(Ross and Ma 2014, p. 79)

In the experiments presented in this paper, we only used cases where prepositional phrases precede verbs.

9   The 15 control filler items were cited from the previous literature on MC, such as Huang et al. (2009), Yao et al. (2022), etc., where the degrees of acceptability of the items (from completely natural to completely degraded) were clearly indicated. Furthermore, we consulted with three native speakers to make sure that the indicated degrees of acceptability were correct. See (i) and (ii) for examples of the fillers: (i) is a completely natural sentence, while (ii) is a completely degraded sentence in MC.

(i)   

| Guanyu | zhe | ge | wenti, | wo | hui | zhijie | gen | Wang | xiansheng | lianxi. |
|--------|-----|-----|--------|-----|-----|--------|-----|------|-----------|---------|
| PREP | this | CLF | matter | I | will | directly | PREP | Wang | mister | contact |

'About this matter, I will directly contact Mr. Wang.'

(ii) *   

| Ni | wei | shenme | fahuo | wo | ne? |
|----|-----|--------|-------|-----|-----|
| you | for | what | get.angry | I | Q |

'Why are you angry with me?'

10   We are including the interaction term in the model because it can help capture complex relationships between variables that may not be apparent when looking at individual effects alone. Assuming a cue-based approach where every cue is equally weighted, we would anticipate an outcome where multiple cues, i.e., the presence of a preposition together with a specific *wh*-phrase, lead to an additive effect. On the other hand, we could assume that one cue when combined with another cue yields an amelioration in the acceptability that goes beyond the contribution of each cue separately: what is known as a super-additive effect. Even though there were no specific theoretical predictions for an interaction, we believe it was a valid approach to investigate whether such effects were in place. Therefore, we decided to include the interaction term as an explorative analysis to determine whether there were moderating effects of one predictor on the relationship with another predictor.

11   It deserves to be mentioned that the average ratings of the exploratory tests presented in footnotes 4 and 7 are somewhat higher than the average ratings of the conditions in our formal experiment. We note that the ratings shown in the footnotes also exist in the formal experiment, as can be seen from the range covered by the SD. We noticed that the range of ratings in the exploratory test also showed inter-speaker variation, which partly motivated us to conduct experiments in formal settings with a larger number of participants, critical items, and fillers than that in the exploratory tests.

12   https://pavlovia.org/, accessed on 12 January 2023.

13   In retrospect, including heavy *wh*-phrases, which provide more cues than specific *wh*-phrases, in the experiment could strengthen our conjecture. We leave this matter for further research.

14   We thank an anonymous reviewer for reminding us of the discussions related to *hot* and *cool* languages (Ross 1982; Huang 1984; Huang 1994; Liu 2014). Languages like MC are seen as *cool* languages, where null arguments are allowed under rich discourse and contexts. Languages like English are *hot* languages, where null arguments are not allowed under discourse and contexts. Moreover, languages like Spanish are *medium-hot* languages, where null subjects are allowed because of overt agreement marking. These differences also support the assumption that discourse-related information plays a more significant role in constructions involving the omission of sentence elements in MC than in English and Spanish.

15   Although conditions 3 and 2 received low ratings, the former was significantly more acceptable than the latter, which was unexpected to us. At the moment, we have no clear explanation for this result and, thus, have to leave it to our future research.

16   We would like to mention that we do not consider *shi* as a cue under the framework of the cue-based retrieval theory. Our considerations are as follows. The function of a cue is to facilitate the processing of sluicing constructions. As a result, the presence of a cue leads to higher acceptability ratings of the relevant constructions. As discussed in this section, the presence of *shi* does not significantly improve the acceptability ratings of multiple sluicing constructions other than those involving bare *wh*-arguments. As a matter of fact, constructions with two *shi*s sometimes receive the lowest acceptability rating among the tested conditions, as shown in Tables 7 and 15. This influence of *shi* on the acceptability ratings of multiple sluicing contradicts the functions of cues. Moreover, if *shi* were a cue, the acceptability ratings of Conditions 3 and 2 in sub-experiment 1 of Experiment 2 should exhibit no differences because the two conditions include the same number of cues, i.e., they each have one cue. In a word, the functions of *shi* revealed by our experimental data do not conform to the functions of cues. We thank an anonymous reviewer for reminding us of this point.

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
