# Peer review of "An Experimental Investigation of Multiple Sluicing in Mandarin Chinese"

_languages, doi:10.3390/languages8010088_

Round 1

Reviewer 1 Report

This paper presents two sets of grammaticality judgment experiments to examine factors that affect the acceptability of multiple sluicing sentences in Mandarin Chinese, a language that has no case marking. Based on existing literature, the authors identified two factors, namely the presence of preposition and the use of specificity-denoting D-linking wh-phrase, and manipulated these factors the experiments. Experiment 1 showed a main effect of prepositionhood and a main effect of specificity. Experiment 2 consisted of four sub-experiments, in order to qualify the potential influence of shi on acceptability ratings. The results showed that only bare wh-arguments require an obligatory presence of shi, and its ameliorating effect is more pronounced if it precedes the second bare-wh-phrase. The results were best explained by the cue-based retrieval approach to ellipsis.

The paper is clearly written. However, there are a few issues that the authors might want to further clarify.

1.      Theoretical framework

The authors used the cue-based retrieval theory, a memory-based approach or processing-driven account to explain the results of their experiments. It would be clearer if the authors provide a clear definition of ‘cues’. What counts as a cue? Why would a lexical item such as a preposition be considered as a cue in Mandarin Chinese?

What exactly is the nature of ‘shi’ in Mandarin Chinese? Is it possible to consider ‘shi’ as a cue under the framework of the cue-based retrieval theory? If not, then why? That is, if the authors are not willing to posit that ‘shi’ is a cue, then they might want to provide justification. But if the answer is positive, then the design of their Experiment 1 might be subject to further scrutiny. See below (#3).

2.      Predictions of the cue-based retrieval theory

The authors spelled out the predictions that are based on prior research or existing theoretical claims. However, it would have been clearer if the predictions are laid out according to the cue-based retrieval theory, given that the authors used that theory to motivate the experiments and to explain their results.  

3.      Design of Experiment 1

Experiment 1 has a 2-by-2 factorial design, manipulating the presence or absence of Preposition and the presence or absence of Specificity of wh-phrase. But given the presence or absence of ‘shi’ was not consistent across conditions, the design appears not to be fully crossed as the authors meant it to be. That’s one major problem I have with this paper, and is perhaps most likely to be the reason why the authors ran a series of experiments in Section 3.2 to justify this design (specifically, as the authors endeavored to show in the paper, it’s necessary to have ‘shi’ precede a bare wh-argument).

But I wonder if this design can be justified in terms of the number of cues, following the cue-based retrieval theory.

If we assume ‘shi’ is a cue, then the number of cues varies across conditions: 2 cues in Condition 1, 3 cues in Condition 2, 3 cues in Condition 3, and 4 cues in Condition 4. The authors found two main effects, and these main effects can be accounted for by the differences in cues.

Also, in terms of the experimental stimuli, the critical lexical items in the background event/sentence differ across conditions (e.g., ‘person/college student/girl’, or ‘something/project/name/toy’). I understand it’s difficult to keep lexical items constant in the current design, yet it would have strengthened the results if the authors do so.

Also, the template of experimental stimuli (N = 24 in Exp. 1; N=32 in Exp. 2) appear rather similar – if not monotonous, always having words like ‘I just do not know…”, at least based on the examples shown throughout the paper. Please provide a bit more detail when describing the stimuli.

4.      Data analyses & results

The authors described how they built ordinal models in the data analysis (p. 15, lns 552-561). They included interactions in their fixed effects, but I don’t see any mention of such interactions in the prediction section (lns. 539-547). In fact, if an interaction was predicted, in what directions would the authors predict, or will an interaction generate any theoretical implication? Perhaps the authors might clarify these issues in their revision.

The authors plot their rating data of Exp. 1 in figures (and they did provide mean ratings of Exp. 2 in tables), yet I would like to know how individual differences might affect the results. Thus, I would recommend the author report both means and SDs of the rating data in texts. Individual differences appear to be rather large in various sub-experiments of Experiment 2. For instance, from Figure 5, there appears to be an interaction, yet no effects were found.

Some minor points:

1. I appreciate it that the authors mentioned a few times in their paper that multiple sluicing in Mandarin Chinese is a marked construction, as in other languages. But are there peculiar reasons why the ratings appear to vary rather radically for the same type of stimuli? For instance, in Fn. 4, the authors ran an exploratory test with 7 Mandarin-speaking participants. (ia) corresponds to the (1) condition of Experiment 1, i.e., [-P, bare WH]. Yet the rating of the exploratory test (i.e., 4.29/7) appears significantly higher than that of Experiment 1 (i.e., 3.2/7). Note that if one eyeballs the rating data on Figure 1, none of the conditions in Experiment 1 reached 4.1 on the 7-point scale. Why would there be such huge variability in the two tests, while both used nearly identical template of the testing stimuli?  

2. In both experiments, the authors included 72 filler items. Please briefly describe those filler trials in terms of the structural types.

Author Response

We wish to thank you for the review of our manuscript. We found your comments very helpful in strengthening our arguments and making the paper more readable. We have revised the paper to respond to your comments. The changes we have made along with your comments are listed in the attachment.  

Reviewer 2 Report

The paper is clearly written and organised, and presents a good overview of the literature, clear methodology and design description.

Re, the discussion of findings, I find the following explanation not fully convincing, or at least not exhaustively elaborated:

"Furthermore, although our experiment shows that specific wh-phrases improve the acceptability of multiple 842
sluicing in MC, the specificity effect is not observed in English or Spanish (Cortés Rodríguez 2021; 2023). 843

We argue tentatively that this difference can be explained by MC being a discourse-oriented language (Huang 844

1984) while English and Spanish are sentence-oriented languages (Wakabayashi 2002). In a discourse-ori- 845

ented language, cues provided by discourse-linked wh-phrases could play an important role in facilitating the 846

processing of multiple sluicing sentences.10 In contrast, cues provided by discourse-linked wh-phrases may 847

not significantly affect the processing of multiple sluicing in sentence-oriented languages.
"

I suggest examining the literature on anaphora, coreference tracking, disambiguation in general in Chinese (eg Huang Yan 1994, Huang Shuanfan 2005) or hot/cold languages, in any case scholarship that is more recent than Huang 1984.

I would suggest that glosses follow the Leipzig glossing rules.

Author Response

Thank you very much for your review of our manuscript. We found your comments very helpful in clarifying our arguments and making the paper more readable. We have revised the paper to respond to your comments. The changes we have made along with your comments are listed in the attachment.
